# Recovery of Li and Co in Waste Lithium Cobalt Oxide-Based Battery Using H_1.6_Mn_1.6_O_4_

**DOI:** 10.3390/molecules28093737

**Published:** 2023-04-26

**Authors:** Hua Wang, Guangzhou Chen, Lijie Mo, Guoqiang Wu, Xinyue Deng, Rong Cui

**Affiliations:** 1Anhui Key Laboratory of Water Pollution Control and Waste Water Recycling, Anhui Jianzhu University, Hefei 230601, China; 2Anhui Key Laboratory of Environmental Pollution Control and Waste Resource Utilization, Anhui Jianzhu University, Hefei 230601, China; 3Anhui Research Academy of Ecological Civilization, Anhui Jianzhu University, Hefei 230601, China; 4School of Environment and Energy Engineering, Anhui Jianzhu University, Hefei 230601, China

**Keywords:** waste lithium cobalt oxide based battery, H_1.6_Mn_1.6_O_4_, citric acid, tartaric acid, regenerated lithium cobaltate

## Abstract

H_1.6_Mn_1.6_O_4_ lithium-ion screen adsorbents were synthesized by soft chemical synthesis and solid phase calcination and then applied to the recovery of metal Li and Co from waste cathode materials of a lithium cobalt oxide-based battery. The leaching experiments of cobalt and lithium from cathode materials by a citrate hydrogen peroxide system and tartaric acid system were investigated. The experimental results showed that under the citrate hydrogen peroxide system, when the temperature was 90 °C, the rotation speed was 600 r·min^−1^ and the solid–liquid ratio was 10 g·1 L^−1^, the leaching rate of Co and Li could reach 86.21% and 96.9%, respectively. Under the tartaric acid system, the leaching rates of Co and Li were 90.34% and 92.47%, respectively, under the previous operating conditions. The adsorption results of the lithium-ion screen showed that the adsorbents were highly selective for Li^+^, and the maximum adsorption capacities were 38.05 mg·g^−1^. In the process of lithium removal, the dissolution rate of lithium was about 91%, and the results of multiple cycles showed that the stability of the adsorbent was high. The recovery results showed that the purity of LiCl, Li_2_CO_3_ and CoCl_2_ crystals could reach 93%, 99.59% and 87.9%, respectively. LiCoO_2_ was regenerated by the sol–gel method. XRD results showed that the regenerated LiCoO_2_ had the advantages of higher crystallinity and less impurity.

## 1. Introduction

With the wide application of lithium batteries in mobile communications, electric vehicles and other fields, environmental problems caused by their e-wastes are becoming increasingly prominent, especially heavy metal pollution caused by cobalt, lithium and manganese in batteries [1]. Therefore, the recovery of valuable metals in waste lithium-ion batteries has attracted the attention of many scholars. The main recovery methods are pyrometallurgy [2] and hydrometallurgy [3]: (1) the first process was simple, but it had the disadvantages of high energy consumption, complex temperature control and easy-to-produce harmful dust and gas; (2) the second process had a high recovery rate of valuable metals and high purity of products, which is the mainstream technology of waste lithium-ion battery recovery at present. The wet process was mainly acid leaching, in which metal ions were dissolved by acid and then selectively recovered from metals such as lithium and cobalt. Among them, common acids could be divided into inorganic acids and organic acids. Inorganic acids were usually hydrochloric acid [4], nitric acid [5], sulfuric acid [6], etc. The above strong acids were used to destroy the structure of positive electrode materials and achieved the purpose of recycling. However, strong inorganic acids had defects, such as corroding machinery and producing waste gas and wastewater. On the contrary, organic acids had mild acidity, little corrosion and were easy to decompose, so they had an excellent performance as leaching agents [7]. In addition, some alkaline leaching processes were used to recover aluminum in batteries [8].

For metal ions after leaching, the precipitation method [9], solvent extraction method [10] and other methods are usually used for separation. Due to the many separation steps and difficult separation, the applicability was poor. Among the treatment methods of heavy metal pollution in water, the adsorption method was often used. For example, microplastics [11] and chitosan [12] had a strong adsorption capacity for metal ions. At present, there is little research on metal ion separation by the adsorption method, the reason being that the common adsorbents often could not adsorb a single metal ion alone. A MnO_2_·0.5H_2_O lithium-ion sieve obtained by elution of a Li_1.6_Mn_1.6_O_4_ precursor is currently the adsorbent with the largest adsorption capacity of lithium ions [13]. Moreover, the Li-ion sieve was highly selective in the adsorption of Li^+^, which provided a possibility for the adsorption method to be applied to the separation of lithium in lithium cobaltate batteries.

In this study, citric acid and tartaric acid in organic acids were selected as acids in the leaching agent, and the leaching experiment was carried out on the waste cathode material of a lithium cobalt oxide-based battery. A lithium-ion screen with selective adsorption capacity was used to adsorb, separate and recover Li^+^ from the leaching solution. Two routes were tested for cobalt recovery: one was to reprecipitate to form cobalt chloride crystals, and the other was to use recycled lithium as a lithium source to regenerate lithium cobaltate.

## 2. Results and Discussion

### 2.1. Leaching Rules of Positive Grade Materials

#### 2.1.1. Powder Parameters of Cathode Material

The material came from the cathode material of a waste lithium cobalt oxide-based battery, which was mainly composed of lithium cobaltate. During the disassembly process, the aluminum shell was manually removed to fully remove the attached organic plastics and adhesives. The element composition of the cathode material was determined, as shown in Table 1. The main metals were 60.2% Co and 7.2% Li, respectively, while other metal elements such as Fe, Al and Ni were all lower than 1%. Lithium cobalt oxide-based batteries on the market might also contain Na, Cu, Zn and other elements, which were not detected in the powder.

#### 2.1.2. Leaching Kinetics

The leaching efficiency of the valuable metals was affected by many factors, such as leaching acid concentration, H_2_O_2_ concentration, temperature, rotational speed and reaction time. To control these factors, the authors conducted a lot of exploration, which referred to other studies in the literature [14,15,16,17]. So, in this study, the temperature was determined to be 90 °C, and the rotation speed was 600 r·min^−1^. With the increase in reaction time, the metal leaching efficiency in the cathode material is shown in Figure 1, where the leaching rates of Co and Li reached the highest of 86.21% and 96.9% around 4 h (Figure 1a). Then, the data were fitted according to the control velocity equation of the chemical reaction (Figure 1b), where X is the leaching efficiency (%) and t is the leaching time (min). The determination coefficients R^2^ of Co and Li fitting curves were 0.95334 and 0.99447, both above 0.95, indicating a high degree of fitting; that is, metal leaching was controlled by chemical reaction.

When tartaric acid was used as the leaching agent, the metal leaching efficiency in the cathode material was shown in Figure 2. The leaching rates of Co and Li basically increased with the increase of time and reached the highest values of 90.34% and 92.47% at about 5 h, respectively. However, they decreased when the time exceeded 5 h. Then, the data were fitted according to the control velocity equation of the chemical reaction. The determination coefficients R^2^ of the fitting curves of Co and Li were 0.99027 and 0.7692, and the fitting curves of Co had a higher fitting degree, which was controlled by chemical reaction, while Li was obviously easier to leach in tartaric acid. The leaching rate reached 75.29% in the first 30 min. The fitting result of the control velocity equation of the chemical reaction was poor, which was not suitable for Li leaching in a tartaric acid system.

Cobalt trivalent could not be directly dissolved by acid, so hydrogen peroxide was used as a reducing agent to make cobalt oxide accept an electron of hydrogen peroxide, then was dissolved by citric acid to form cobalt citrate. The leaching reaction is described by Equation (1) [18].
(1)$$2H_{3}\mathrm{Cit}\left(\mathrm{aq}\right)+2{\mathrm{LiCoO}}_{2}\left(s\right)+{\;H}_{2}O_{2}\left(\mathrm{aq}\right)=2{\mathrm{Li}}^{+}\;\left(\mathrm{aq}\right)+2{\mathrm{Cit}}^{3-}\;\left(\mathrm{aq}\right)+2{\mathrm{Co}}^{2+}\left(\mathrm{aq}\right)+4H_{2}O\;+{\;O}_{2}\left(g\right)$$

When tartaric acid was used as the leaching solution, it also played a reducing role. So, the involvement of hydrogen peroxide was not required. The reaction with lithium cobaltate is described by Equation (2) [7].
(2)$${\mathrm{LiCoO}}_{2}+3C_{4}H_{6}O_{6}\to \mathrm{Co}{\left(C_{4}H_{5}O_{6}\right)}_{2}+{\mathrm{LiC}}_{4}H_{5}O_{6}+2H_{2}O$$

According to the chemical equation, the complete reaction between citric acid and lithium cobaltate should be 1:1, but in the actual process, only excessive citric acid could ensure the complete leaching of metal in lithium cobaltate material. As shown in Figure 3a, when the ratio of solid to liquid in the reaction system was low, the cobalt lithium element could be well leached. However, when the ratio of solid to liquid increased, it became difficult for the products on the surface of the material to diffuse into the solution with higher concentration, so the efficiency decreased. According to the influence of the solid–liquid ratio on the leaching rate, the best solid–liquid ratio was 10 g·1 L^−1^.

In the same way, an excessive ratio of tartaric acid to lithium cobaltate could be used for complete leaching. When the solid–liquid ratio was 20 g·1 L^−1^, the best leaching rates of Co and Li were the highest, reaching 91.86% and 93.66%, respectively, which was similar to the citric acid system. When the solid–liquid ratio increased, there was also a decline in the tartaric acid system. When the solid–liquid ratio increased, it became difficult for the products on the material surface to diffuse to the solution with a larger concentration.

The cathode material powder before and after leaching was characterized by XRD, as shown in Figure 4. The main composition of the cathode material in the figure was LiCoO_2_ (PDF#75-0532). Compared with the XRD pattern of the leached material, it was obvious that the characteristic diffraction peaks belonging to LiCoO_2_ basically disappeared, and only some stray peaks with poor matching degree were left, indicating that the structure of lithium cobaltate was destroyed in the leaching process, and the valuable metal was fully dissolved in the leaching solution.

### 2.2. Preparation of Lithium Ion Sieve

The precursor of the Li_1.6_Mn_1.6_O_4_ lithium-ion sieve was difficult to obtain by direct treatment of lithium manganese compounds with a traditional solid phase reaction in one step, so in this paper, a Li_1.6_Mn_1.6_O_4_ lithium-ion sieve was prepared step-by-step using the sol–gel method, hydrothermal method and solid phase calcination method. The sol–gel method was used to first add LiOH aqueous solution to MnCl_2_ solution; at this time, Mn^2+^ would generate Mn(OH)_2_ precipitation under alkaline conditions. Then, 30% H_2_O_2_ was slowly added, making H_2_O_2_ and Mn(OH)_2_ react to γ-MnOOH; dropping speed was very important. If H_2_O_2_ drops were added too quickly, part of Mn(OH)_2_ precipitates would be oxidized directly into MnO_2_, making γ-MnOOH precipitates contain a large amount of MnO_2_, thus increasing the impurity of the target product.

The γ-MnOOH reacted to the lithium source of the above reactor by hydrothermal treatment, then formed LiMnO_2_. Finally, the precursor of Li_1.6_Mn_1.6_O_4_ was obtained by the calcination of LiMnO_2_ [19].

The precursor of the lithium-ion screen needed elution before adsorption. Firstly, Li^+^ was stripped from the precursor by the eluent, that is, through the H–Li-ion exchange process. A void filled with H^+^ was formed in the material, and the eluted void played a role in screening and memory of Li^+^. The XRD of the prepared lithium-ion before and after sieve elution is shown in Figure 5. As seen from Figure 5, the diffraction peaks appeared at 2θ = 19.1°, 37.0°, 45.0° and 65.5°, respectively, corresponding to the (111), (311), (400) and (440) crystal planes, and there were basically no mixed peaks. The XRD characteristic peaks did not change significantly before and after pickling, except that the cells would become slightly smaller. The reason for this was that during the elution, H^+^ was the only ion exchanged with Li^+^ in Li_1.6_Mn_1.6_O_4_, so its basic structure did not change [20]. Additionally, the XRD results could prove that our prepared Li_1.6_Mn_1.6_O_4_ was highly stable and would not be destroyed by acid washing.

The infrared spectra of the above materials before and after adsorption are shown in Figure 6. It can be seen that the absorption peaks at 3424.7 cm^−1^ and 1593.1 cm^−1^ are related to the vibration peaks of O-H in the adsorbed water. The absorption peak at 1112.7 cm^−1^ is the stretching vibration peak of the Li–O bond. The absorption peaks at 619.1 cm^−1^ and 528.2 cm^−1^ are related to Mn(IV)-O and Mn(III)-O vibration peaks, respectively [21,22]. By comparison, no matter whether before the pickling, before the adsorption or after the adsorption of Li^+^ ions, the infrared spectrum of Li_1.6_Mn_1.6_O_4_ had almost no change, indicating that the adsorption mechanism of Li_1.6_Mn_1.6_O_4_ did not contain chemical bond breaking; its main mechanism is generally believed to be ion exchange [23] and surface disproportionation [24].

### 2.3. Lithium Recovery

#### 2.3.1. Pretreatment of Eluent and Selection of Optimum Factors

The main metal ion components of the leaching solution were Co^2+^ and Li^+^, and the detected contents were 5.862 g·L^−1^ and 0.775 g·L^−1^, respectively, except for a very small amount of Fe^2+^, Ni^2+^ and Al^3+^. The solution was acidic; however, the acidic condition was not conducive to the adsorption of the lithium-ion sieve. In order to eliminate the interference of other factors, a control group was set up in the experiment, and 2 g of lithium-ion sieve powder was taken to adsorb 100 mL 10 g·L^−1^ lithium solution, as shown in Figure 7, which showed the changes of ion sieve adsorption capacity under different pH conditions. It can be seen that when the pH of the solution was low, the concentration of H^+^ in the solution was high, resulting in the ion exchange being difficult to occur. On the contrary, the lower concentration of H^+^ promoted ion exchange.

Therefore, the pH of the leaching solution was adjusted by adding an appropriate amount of ammonia water, and the pH of the leaching solution was 9.5 when Fe(OH)_2_ was completely precipitated (the residual ion concentration was less than 10^−6^ mol·L^−1^). Al(OH)_3_ is 6, Ni(OH)_2_ is 8.4 and Co(OH)_2_ began to precipitate above 9 [25]. Therefore, the pH was adjusted to about 9–10, and impurity ions in the leaching solution could hardly be detected by filtering the sediment. The contents of Co^2+^ and Li^+^ in the leaching solution were 5.316 g·L^−1^ and 0.739 g·L^−1^, respectively. The concentration of the solution was closely related to the adsorption capacity of the adsorbent. Figure 7 showed that with the increase in concentration, the sieve adsorption capacity of lithium-ion increased from 19.97 mg⋅g^−1^ to 34.76 mg·g^−1^. This was because the high concentration of Li^+^ could promote the H–Li ion exchange process, thus increasing the adsorption capacity. Therefore, the leaching solution could be heated to concentrate, in order to facilitate the adsorption of the lithium-ion sieve.

#### 2.3.2. Lithium-Ion Sieve Adsorption

An amount of 100 mL concentrated leachate, including the concentration of Li^+^ of about 1.8 to 2 g L^−1^, was added to 2 g lithium-ion sieve adsorption. Its adsorption performance is shown in Figure 8. It can be seen that the lithium-ion sieve and the leaching solution reached the adsorption equilibrium for about 48 h. Under this condition, the maximum adsorption capacity was 38.05 mg·g^−1^, accounting for 98.21% of the total Li^+^ content. Furthermore, the quasi-first-order adsorption kinetics model and the quasi-second-order adsorption kinetics model were fitted. The fitting results showed that the first-order kinetic decision coefficient R^2^ of the lithium-ion screen was 0.93542, lower than the second-order kinetic 0.99956, indicating that the adsorption of the lithium-ion screen on the leach solution was more consistent with chemical adsorption. Subsequently, the concentration of Co^2+^ was detected. There was a certain decrease in Co^2+^ concentration, about 0.29% of the total, which was almost negligible. The reason was mainly due to the lithium ion screen having a superior selective adsorption capacity.

Repeated elution experiments of the lithium-ion sieve were carried out, and the results are shown in Table 2. It can be seen that the adsorption capacity of the lithium-ion screen decreased significantly in the first three cycles at 8.4625%, 10% and 6%, and the change slowed down to 3.82% from the fourth to the fifth time, and the final performance of the twentieth time decreased to only 10.46% compared with the tenth time. The results showed that the repeated adsorption performance of the lithium-ion screen was good. When HCl was used as the eluent, the dissolution rate of lithium was about 91% in many cycle experiments, which could achieve a good dissolution effect.

#### 2.3.3. Synthesis of Lithium Chloride

The adsorbed lithium-ion sieve powder was desorbed using 0.5 mol·L^−1^ HCl solution. However, due to the disproportionation of Mn on the surface of the lithium-ion sieve during the elution process, a small amount of Mn^2+^ existed in the solution. The removal effect could be achieved by dropping an appropriate amount of H_2_O_2_ and diluting ammonia water. In addition, the Co^2+^ adsorbed during the adsorption process was not detected in the solution and showed that a trace amount of Co^2+^ on the adsorbent did not eluate during the elution process. Through the above operation, the purer LiCl solution was finally obtained.

The obtained pure LiCl solution was heated, concentrated, cooled and crystallized to obtain some white crystals. Phase analysis was conducted by XRD, as shown in Figure 9. In Figure 9, the diffraction peaks were sharp and basically smooth, and there were almost no mixed peaks. The position of the diffraction peaks was consistent with that of the standard card (PDF#74-1972). It indicated that the obtained product was crystalline compared to pure lithium chloride crystal, and the peak at about 2θ=33° was caused by LiCl·H_2_O. Its purity reached about 93% (determined by the Li^+^ content in solids).

#### 2.3.4. Synthesis of Lithium Carbonate

Although the synthesis method of lithium chloride crystal was simple, the conversion rate in the process was low, resulting in a lot of waste. Therefore, the lithium recovery method could be designed through the following equation:(3)$$2{\mathrm{Li}}^{+}+{\mathrm{Na}}_{2}{\mathrm{CO}}_{3}=2{\mathrm{Na}}^{+}+{\mathrm{Li}}_{2}{\mathrm{CO}}_{3}↓$$

According to Equation (3), only by controlling the dosage of Na_2_CO_3_ and the reaction temperature can Li_2_CO_3_ precipitation be generated, and then the reaction suspension can be filtered, washed and dried to obtain Li_2_CO_3_ crystals.

The influence of the pH;

As 0.5 mol·L^−1^ HCl was used as eluent, the recovered lithium solution became acidic (pH 0~1). The formation pH of Li_2_CO_3_ crystal was determined by many experiments to be about 8~9, so the solution pH could be adjusted by adding an appropriate amount of NaOH before the reaction to promote the formation of Li_2_CO_3_ crystals. Figure 10 shows a comparison of the quality and purity of the final product when 10 mL of a 30 g·L^−1^ Na_2_CO_3_ solution was added to 100 mL of leachate at different pH values (5, 6, 7, 8, 9, 10, 11 and 12). It can be seen that as the pH of the initial solution rose, the crystal yield of Li_2_CO_3_ gradually increased because a small amount of Na_2_CO_3_ reacted with H^+^ when the solution was acidic. The purity of the product tended to rise at the beginning. The closer the pH was to 9, the easier Na_2_CO_3_ would react to form crystals. When the pH exceeded 9, the purity began to decline. Therefore, the pH of the recovered lithium solution could be adjusted to 7~9, which is conducive to the generation of high quality Li_2_CO_3_ crystals.

2.The influence of the dosage of Na_2_CO_3_;

The dosage of Na_2_CO_3_ was changed, and different volumes of 30 g·L^−1^ Na_2_CO_3_ solution (5, 7, 10, 15 and 20 mL) were added into 100 mL of eluent, respectively. Other parameters were fixed unchanged, and the experiment was conducted; the results are shown in Figure 11. As seen in Figure 11, with the increase in the dosage of Na_2_CO_3_, more crystals of Li_2_CO_3_ can be obtained. When the dosage of Na_2_CO_3_ was greater than 15 mL, without regard to the loss in the filtration process, the mass increase of the obtained Li_2_CO_3_ crystals was small because the Li in the eluent had almost completely formed precipitate. However, the crystallization purity of Li_2_CO_3_ tended to decrease slightly with the increase of the dosage of Na_2_CO_3_ because a small amount of sodium carbonate was wrapped in the Li_2_CO_3_ crystal during the crystallization process [26].

3.The influence of reaction temperature.

Temperature was also a key factor in the formation of Li_2_CO_3_ precipitation. Other conditions remained unchanged, and the temperature was controlled at 20, 30, 40, 60 and 80 °C for the experiments. The results are shown in Figure 12. It can be seen from Figure 12 that the crystal weight of Li_2_CO_3_ increased with the increase in temperature. The solubility of Li_2_CO_3_ decreased with the increase in temperature. According to the study of Tao et al. [26], the Li_2_CO_3_ aggregates generated at low temperature were mainly flake, while the Li_2_CO_3_ aggregates generated at high temperature were mainly rod-like. Compared with rod-like aggregates, flake aggregates were more likely to contain impurities, which was also the reason why the product purity was lower at low temperature.

For 100 mL of the treated lithium-containing solution (Li concentration is about 40~50 mg·L^−1^), considering the influence of the dosage of Na_2_CO_3_ and the reaction temperature, using 15 mL Na_2_CO_3_ solution with a concentration of 30 g·L^−1^ and a control temperature of 60 °C could make the production of Li_2_CO_3_ larger and the purity higher.

### 2.4. Recovery of Cobalt

Two basic routes for cobalt recovery were designed: one was to regenerate lithium cobaltate by leaching solution after adsorption, and the other was to extract pure cobalt chloride crystals by precipitation. The former avoided the use of a large number of chemical reagents but added the steps of vacuum-drying and high-temperature calcination. The latter was simpler to operate but used more chemical reagents.

#### 2.4.1. Regeneration of Lithium Cobalt Oxide-Based Battery by Soft Chemical Method

The element concentration ratio was adjusted to 1:1 according to the concentration of Co^2+^ in the leaching solution. The corresponding lithium chloride crystals obtained in the above steps were added to the leaching solution as lithium sources; then, the pH of the leaching solution was adjusted to 7 with ammonia. Firstly, this was placed in a water bath kettle at 80 °C, then magnetically stirred for about 2 h at 200 r·min^−1^ until the formation of sol. Subsequently, the sol formed a dry gel through vacuum drying for 24 h, and then the calcination was carried out by a tubular furnace at 700 °C temperature to obtain the regenerated LiCoO_2_ crystal.

The calcination temperature had a great influence on the performance and structure of the regenerated LiCoO_2_. In summary of the studies [27,28,29], when the temperature was 500 °C, the regenerated LiCoO_2_ was mostly a spinel structure, and when it rose to between 700 °C and 900 °C, the regenerated LiCoO_2_ was a hexagonal phase structure. With the increase in temperature, the peak strength would gradually increase and the crystallinity would increase. However, when the temperature exceeded 900 °C, lithium ions would gradually come out and decompose to produce Co_3_O_4_. Therefore, this study chose the calcination condition at 700 °C for 5 h to obtain regenerated LiCoO_2_ particles. Phase analysis by XRD showed that the position of the diffraction peak was consistent with that of the standard card (PDF#74-1972), with the characteristic diffraction peaks of (003), (101) and (104), except that there were two LiCoO_2_ characteristic peaks of hexagonal layered structures (018) and (110) [28,29], indicating that LiCoO_2_ material with a better structure was obtained using the sol–gel method (Figure 13). SEM images of the regenerated lithium cobaltate are shown in Figure 14. Seen from Figure 14, the crystals of lithium cobaltate were obvious.

#### 2.4.2. Synthesis of Cobalt Chloride

Due to the excessive amount of citric acid added in the leaching experiment, large amounts of citric acid crystals might be generated if crystallization is directly evaporated. Therefore, a solution with a concentration of 5 mol·L^−1^ NaOH was slowly dripped in and filtered when the solution produced a large amount of blue–gray precipitate, and then a precipitate of coarse cobalt hydroxide was obtained. Hydrochloric acid was used for dissolution, then filtered, evaporated and crystallized for the remaining solution. Finally, relatively pure CoCl_2_ was obtained. As shown in Figure 15, the position of the diffraction peak of the prepared CoCl_2_ was consistent with that of standard card (PDF#25-0242), indicating that its crystallinity was good, and the crystal purity was about 87.9% after multiple tests.

## 3. Materials and Methods

### 3.1. Reagents and Instruments

Main reagents: lithium hydroxide monohydrate, citric acid, tartaric acid, hydrogen peroxide (Shanghai Maclin Biochemical Technology Co., Ltd., Shanghai, China), manganese chloride tetrahydrate, sodium hydroxide, hydrochloric acid and ammonia (Sinopharm chemical reagent Co., Ltd., Shanghai, China). In this study, the raw material was the fine particles obtained by the discharge, disassembling, crushing and screening of a waste lithium cobalt oxide-based battery whose surface binder and acetylene ash were removed by calcination. The experimental water was deionized water.

Main instruments: UV-26001 UV-visible spectrophotometer (Shimadzu Company, Kyoto, Japan), Nicolet 380 Fourier Transform infrared spectrometer (Thermo Fisher, Waltham, MA, USA), Regulus8100 Scanning electron Microscope ( HITACHI, Ibaraki, Japan), X-ray Photoelectron Spectrometer (Thermo Fischer, Waltham, MA, USA), Optima 8000 Inductively Coupled Plasma Spectrometer (PERKINELMER, Waltham, MA, USA), Tube furnace.

### 3.2. Experimental Process

Leaching of valuable metals: Weighed 0.3 mol citric acid (tartaric acid) and dissolved in 200 mL deionized water, followed by ultrasonic shock for 5 min. After that, appropriate amount of hydrogen peroxide solution was added (tartaric acid does not require hydrogen peroxide), weighed 2, 4, 6, 8 and 10 g of treated waste battery positive powder and added it to the beaker. The beaker was placed in a water bath kettle at the temperature of 90 °C and a rotating speed of 600 r·min^−1^. ICP-AES was used to determine the metal concentration in the leaching solution, and the metal leaching rate was calculated as shown in Equation (4):(4)$$\eta =\frac{c\times V}{m}\times 100\%$$

In the formula: *η* is metal leaching rate, %; *c* is the mass concentration of metal ions, g/mL; *V* is filtrate volume, mL; *m* is the theoretical mass of each metal in lithium cobaltate powder, g.

Preparation of the adsorbent: Took 0.1 mol MnCl_2_·4H_2_O and dissolved it in 100 mL deionized water, added a quantitative amount of LiOH·H_2_O (molar ratio of lithium to manganese was 4:1) and mixed evenly. After mixing, slowly added appropriate amount of 30% H_2_O_2_, about 5 mL, continued to stir for 2 h, and then aged for 24 h. Transferred the formed colloid to a 100 mL Teflon-lined high-pressure reactor by rinsing with supernatant, then added partial supernatant to about 70% of the autoclave volume and heated at 120 °C for 24 h. The material obtained was filtered, washed, dried, then ground with mortar and calcined in Muffle furnace at 400 °C for 5 h to get the precursor of ion sieve. Weighed 10 g lithium-ion sieve precursor powder and placed it in 0.5 mol·L^−1^ HCl solution, stirred it with 200 r·min^−1^ magnetic force for about 12 h, and then obtained lithium-ion sieve adsorbent through filtration, washing and drying process.

Separation of metal ions: Put 2 g ion sieve powder into 100 mL leach solution, stirred at about 20 °C with 200 r·min^−1^ magnetic force for 12 h, sampled at selected time period, measured Li^+^ concentration, and drew saturation adsorption curve to study the adsorption performance of lithium-ion sieve. Then, the solution was filtered, lithium-ion sieve powder was collected, and the adsorbed lithium ion sieve was placed in 0.5 mol·L^−1^ HCl solution for desorption to obtain a higher purity lithium solution.

## 4. Conclusions

The leaching of the cathode material of a lithium cobalt oxide-based battery with citric acid and a hydrogen peroxide system was investigated. The leaching rates of 86.21% and 96.9% for Co and Li can be achieved at a reaction temperature of 90 °C, stirring speed of 600 r·min^−1^ and a solid–liquid ratio of 10 g·1 L^−1^. The determination coefficients R^2^ of Co and Li fitting curves were 0.95334 and 0.99447 according to the chemical reaction control velocity equation, indicating that leaching was a chemical reaction. In the tartaric acid system, the leaching rates of Co and Li were 90.34% and 92.47%. The leaching of Li in the tartaric acid system was rapid, reaching 75.29% in the first 30 min;The prepared Li_1.6_Mn_1.6_O_4_ lithium-ion screen was used to conduct the lithium separation. The maximum adsorption capacity of the lithium screen was 38.05 mg·g^−1^, and the dissolution rate of lithium was about 91%. Through elution, purification and other steps, the adsorbed lithium was transformed into a lithium chloride solution. There were two methods for lithium recovery: (1) Relatively pure lithium chloride crystals can be obtained by direct concentration and crystallization, with a detected purity up to 93%. (2) Li_2_CO_3_ crystals were generated by adding Na_2_CO_3_, and the purity of Li_2_CO_3_ crystals can reach 99.59%;Two methods were used to conduct the tests for the recovery of cobalt: (1) CoCl_2_ crystals were obtained by means of reprecipitation and recrystallization, and the purity can reach 87.9%; (2) LiCoO_2_ was directly regenerated using the sol–gel method and lithium chloride was used as a lithium source. XRD characterization showed that LiCoO_2_ had good crystallinity. The two methods had different advantages and disadvantages: the former was simpler to operate, and the latter used fewer chemical reagents. The above two methods could provide ideas for recycling waste cathode materials of lithium cobalt oxide-based batteries.

## Figures and Tables

**Figure 1 molecules-28-03737-f001:**
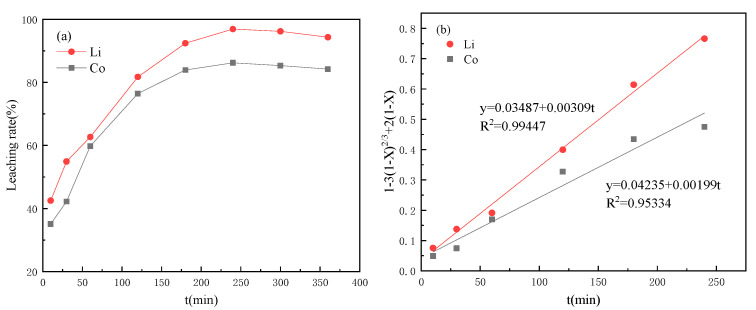
Curves of citric acid leaching performance (**a**) curves with time; (**b**) dynamic fitting curves.

**Figure 2 molecules-28-03737-f002:**
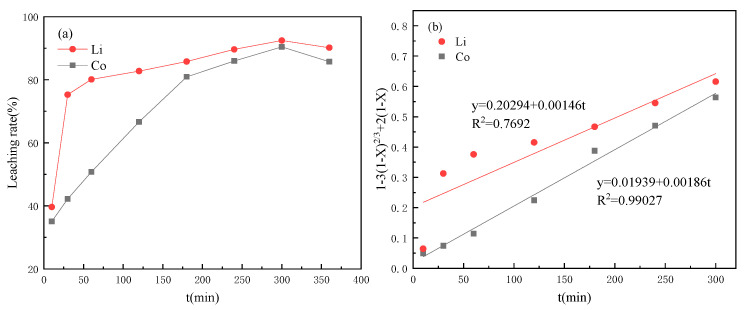
Leaching performance curves under tartaric acid (**a**) change curves with time; (**b**) dynamic fitting curves.

**Figure 3 molecules-28-03737-f003:**
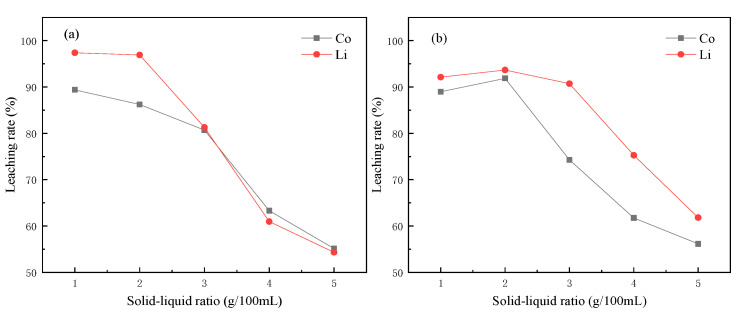
Influence of solid–liquid ratio on leaching rate (**a**) citric acid; (**b**) tartaric acid.

**Figure 4 molecules-28-03737-f004:**
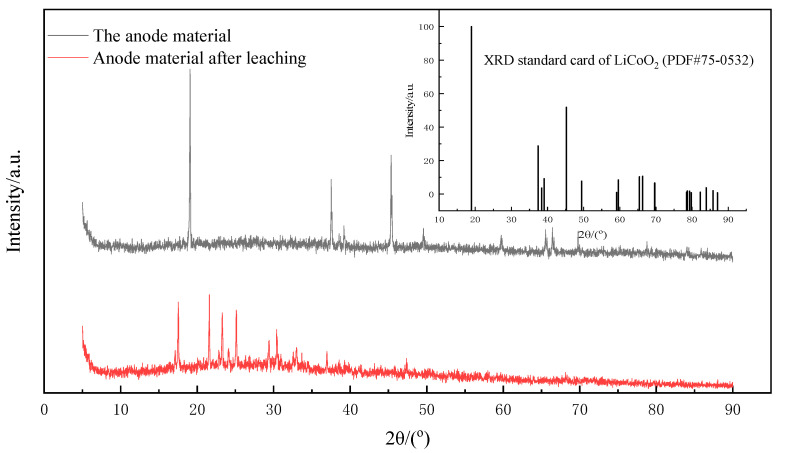
XRD images of the cathode material before and after leaching.

**Figure 5 molecules-28-03737-f005:**
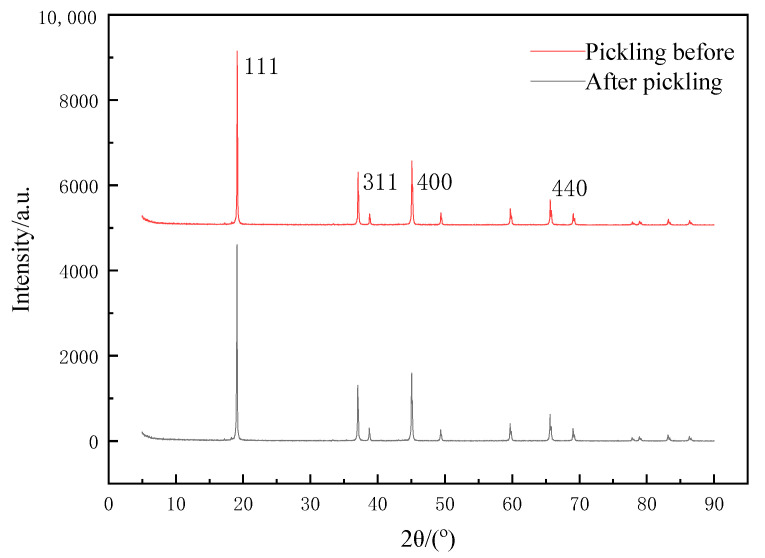
XRD pattern of lithium-ion sieve before and after pickling.

**Figure 6 molecules-28-03737-f006:**
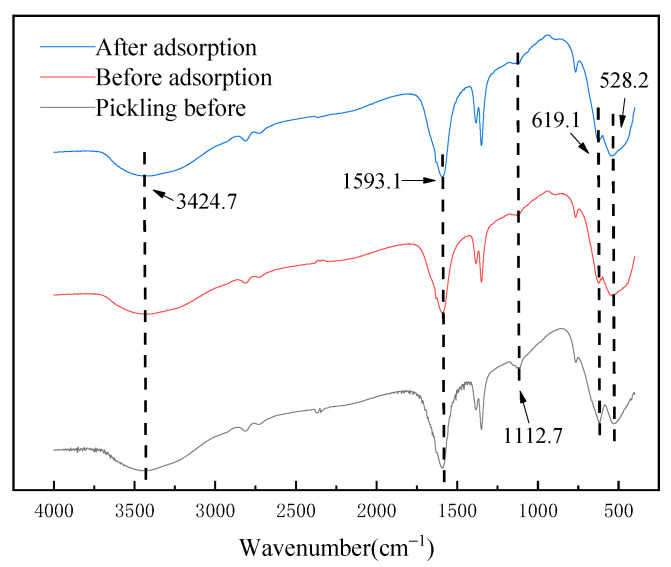
Infrared characterization of lithium-ion sieves before acid pickling and before and after adsorption.

**Figure 7 molecules-28-03737-f007:**
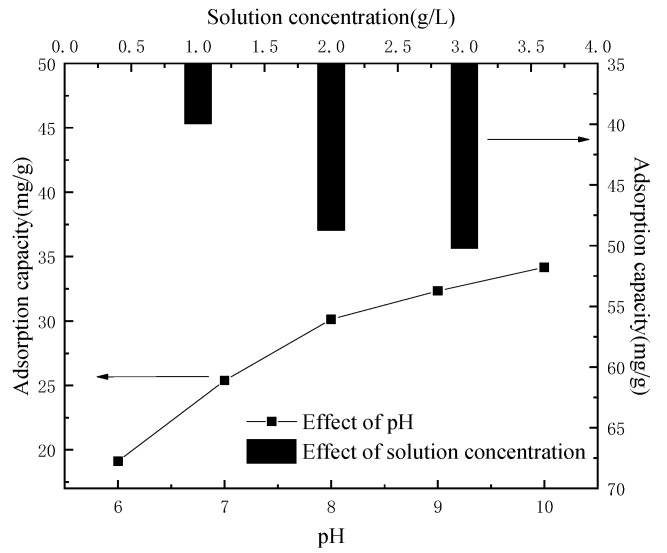
Adsorption capacity in relation to pH and solution concentration.

**Figure 8 molecules-28-03737-f008:**
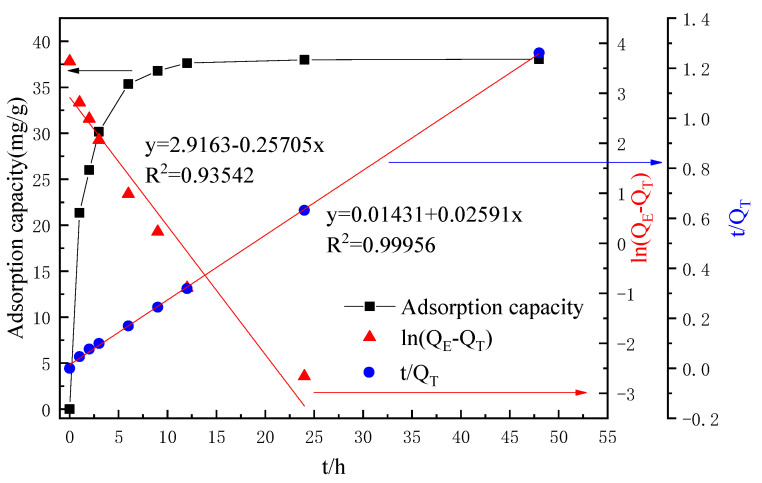
Adsorption performance curves.

**Figure 9 molecules-28-03737-f009:**
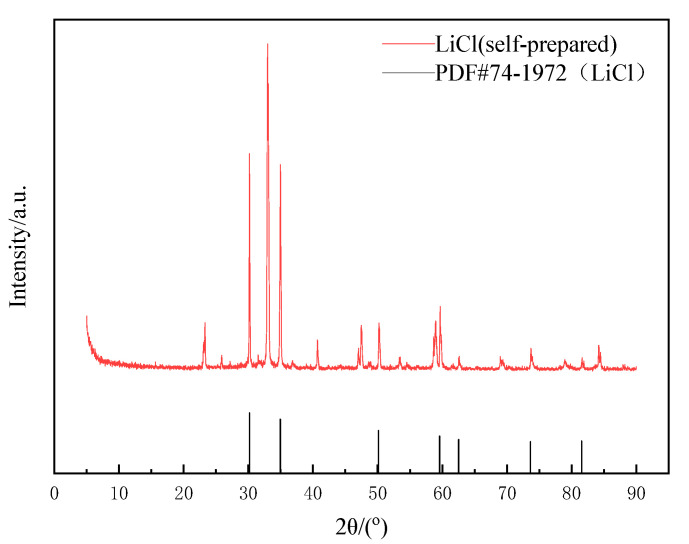
XRD pattern of lithium chloride crystal.

**Figure 10 molecules-28-03737-f010:**
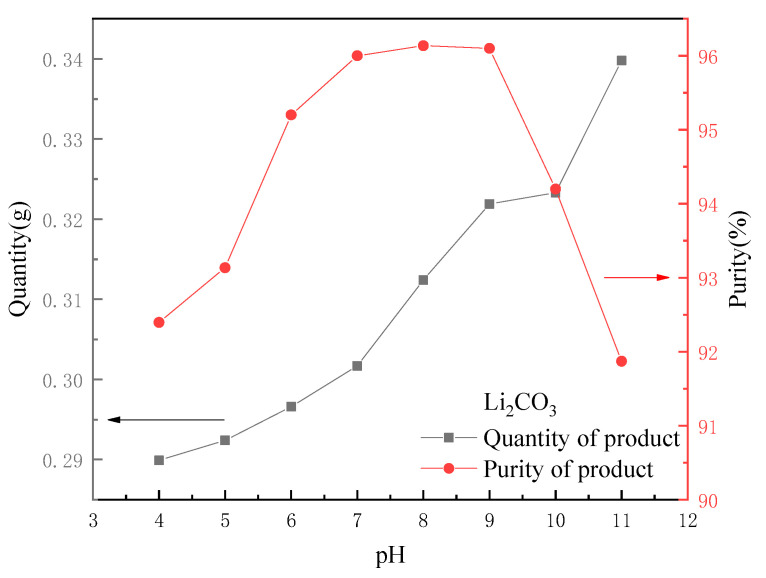
Effect of pH on quantity and purity of output.

**Figure 11 molecules-28-03737-f011:**
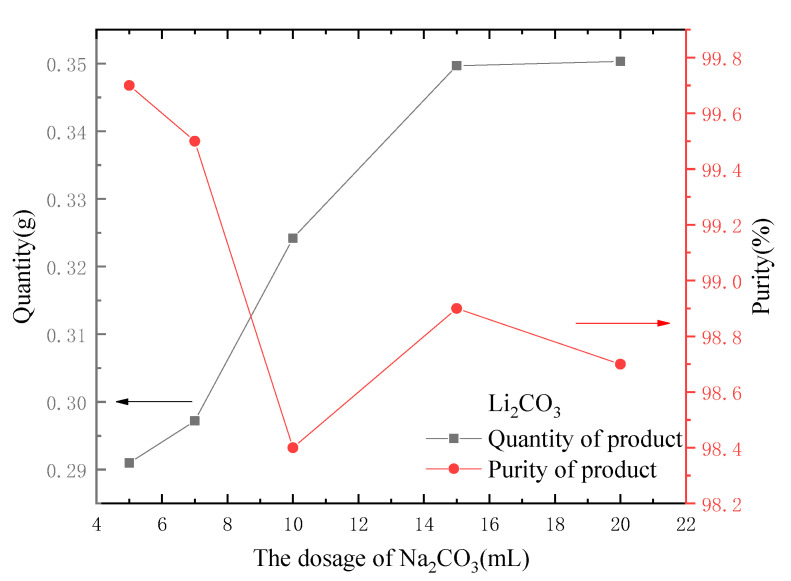
Effect of dosage of Na_2_CO_3_ on the quantity and purity of output.

**Figure 12 molecules-28-03737-f012:**
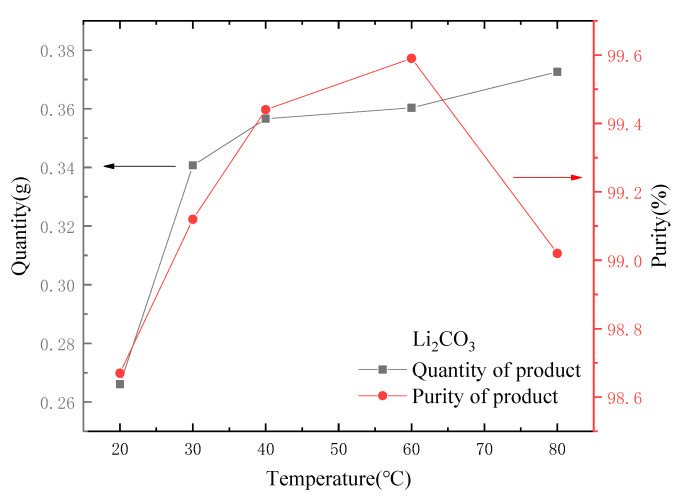
Effect of temperature on quantity and purity of output.

**Figure 13 molecules-28-03737-f013:**
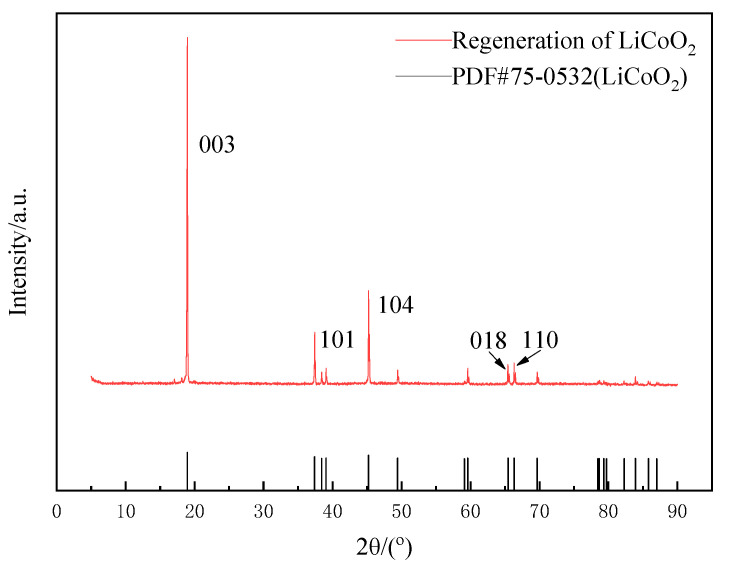
XRD pattern of regenerated lithium cobaltate.

**Figure 14 molecules-28-03737-f014:**
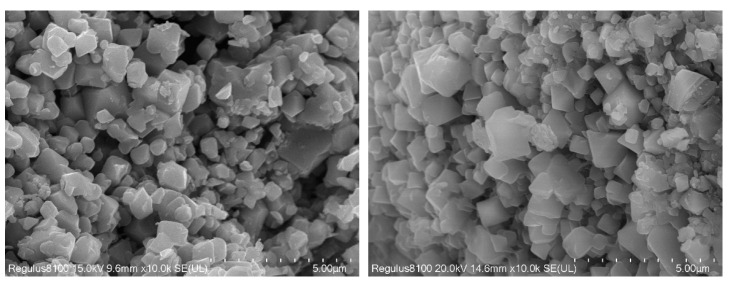
SEM images of regenerated lithium cobaltate.

**Figure 15 molecules-28-03737-f015:**
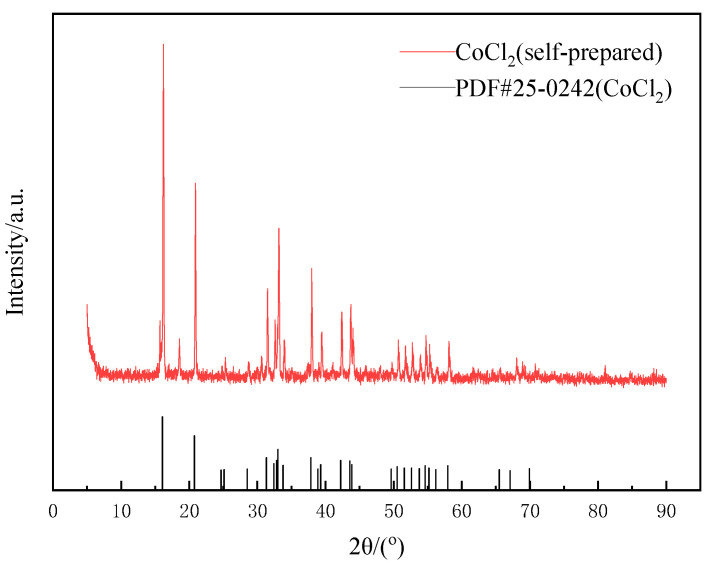
XRD pattern of cobalt crystallization.

**Table 1 molecules-28-03737-t001:** Positive material group division table.

Type of Metal	Li	Co	Fe	Al	Ni
**Proportion (%)**	7.2	60.2	0.02	0.01	0.01

**Table 2 molecules-28-03737-t002:** Results of cyclic adsorption experiment.

Number of Cycles	Adsorption Capacity (mg·g^−1^)	The Dissolution Rate of Lithium (%)
1	38.05	91.54
2	34.83	91.49
3	31.34	90.81
4	29.29	91.12
5	28.17	91.29
10	21.79	91.38
20	19.51	89.88

## Data Availability

Not applicable.

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
