# Peer review of "Recovery of Li and Co in Waste Lithium Cobalt Oxide-Based Battery Using H1.6Mn1.6O4"

_molecules, 2023, doi:10.3390/molecules28093737_

Round 1
Reviewer 1 Report
Lithium recovery is an important research content. The paper focuses on the recovery of lithium and cobalt from waste batteries, which has important research significance and value. The content of the paper is relatively complete and clearly explained, but there are still some small issues that need to be further improved. The detailed questions are as follows:
1.Line 19, 600 r·min-1 should be 600 r·min-1
2.Line 83, A certain amount of citric acid (tartaric acid), Ambiguous expression
3. Line 90, in deionized water, What is the volume?
4. Line 92, what is the total volume at this time?
5.Line 112,table 5, what are the other remaining ingredients?
6.Line 122, how is the leaching rates of Co and Li after 4 hours?
7. Line 127, figure 1 (b), what is the meaning of X in the vertical coordinate axis?
8.Line 149, what about the increase of liquid-solid ratio? should check the increase of liquid-solid ratio.
9.Line 232, what is meaning of left vertical axis in Figure 7?
10.Line 282, Na2CO3 should be Na2CO3
11. Line 307, appropriate amount of lithium chloride crystal, Ambiguous expression
Author Response
Dear Reviewer:
On behalf of my co-authors, we are very grateful to you for giving us an opportunity to revise our manuscript. we appreciate you very much for your positive and constructive comments and suggestions.
We have tried our best to revise our manuscript according to the comments. The following are the responses on an item-by-item basis. Thanks again to the hard work !
Revised portion are marked in red in the paper. Due to the modification of the content, the line number has changed. The main corrections in the paper and the point to point responds to the reviewer’s comments are as following.

Reviewer 2 Report
Recovery of Li and Co in the waste lithium cobaltate batteries using H1.6Mn1.6O4
The concept behind the paper is worth investigating. It initially it starts well and then becomes very confusing and difficult to read.
The reviewer recommends a complete rewrite and the authors should get someone with a chemistry/battery recycling background to assist in writing the manuscript. There is significant chemistry in the manuscript, but the scientific language used is not from a chemist discipline and therefore difficult to read, understand the hypothesis and result from each experiment. For example, solid-liquid ratio would be easier described as concentration and soft chemical synthesis means??? It is especially noticeable that a chemist has not been involved in preparing the manuscript when reading the experimental section when no quantities are provided but instead “a certain amount of citric acid was weighed and dissolved in 200 mL water”. The experimental should allow the reader to repeat the experiment as described and obtain the same yields and ratios of products. This is not possible with the experimental provided as there is insufficient information: this needs to be corrected.
“Again the separation of metals ions” section. Appropriate amount of ion sieve was placed in 100 mL of leaching solution” is meaningless without a stated amount. Quantities and amounts need to be provided otherwise others cannot repeat the experiments. Also, there is no mention of the pH of these solution at which the chemistry is undertaken, which will be critical.
Line 143 they indicate that “hydrogen peroxide, the insoluble Co3+ was generated into Co2+ through the following reaction. Hydrogen peroxide is an oxidising agent where it oxidises other materials and it gets reduced. The sentence does not make sense. In addition, in equation 1 2LiCoO2 has an oxidation state of 3+. So it is not clear from the paragraph what is getting oxidised and reduced. From this equation onwards it become very hard to read and understand how some of the values were obtained (i.e. leaching rate – not described in the Experimental process section. I assume by UV/Vis spectroscopy which has not been described at all). Also the preparation of the lithium ion sieve (3.2.1) should be before Lithium recovery section (3.2). This needs to be described in the Experimental section as well and the reader needs to know how it was made and characterised before it is used to recover lithium ions.
Again not clear what adsorption and pickling concepts are as these have not been defined. For instance, Figure 6, the plots look all the same!
After this point the narrative was lost and the reviewer gave up reading any further.
There are also a lot of typos and misuse of words/tense.
The work looks interesting, but the authors have made it very difficult for the reader to understand and therefore are doing themselves a disservice.
I highly recommend getting a chemist or some in the field of LIB recycling to assist in rewriting the manuscript and then resubmit.
Author Response
Dear Reviewer:
On behalf of my co-authors, we are very grateful to you for giving us an opportunity to revise our manuscript. we appreciate you very much for your positive and constructive comments and suggestions.
We have tried our best to revise our manuscript according to the comments. The following are the responses on an item-by-item basis. Thanks again to the hard work !
Revised portion are marked in red in the paper. Due to the modification of the content, the line number has changed. The main corrections in the paper and the point to point responds to the reviewer’s comments are as following appendix.

Reviewer 3 Report
The paper deals with the recovery of Li and Co from waste LiCoO2 battery. The paper may be accepted after addressing the following corrections:
1. Line 15-21: The sentence is too long. Should be split into two.
2. Line 32: “The introduction”: repetition.
3. Line 37-41: The sentence is confusing
4. Line 72-73: Manganese chloride tetrachloride: It is Manganese chloride tetrahydrate?
5. Line 89: correct spelling-“determinated”
6. Line 192-193: It is mentioned that Mn(OH)2 is formed by the heating MnOOH. Is it correct? Normally, MnO2 will be formed. Please check
7. Line 199-201: the sentence is confusing..
8. Table 2: The values reported in Table 2 are not matching with the description in the text, particularly the values.
9. Figure 8: Please check the caption. It is given as XRD pattern of lithium crystallization.
10. Line 328-329: How the crystallinity is confirmed from SEM?
11. Line 336-338: rewrite sentence
12. Few places, it is given lithium cobalt acid battery. This should be corrected as lithium cobalt oxide based battery
13. Line 355: add respectively.
14. The results are discussed in past tense, especially while mentioning Figures, table etc and also for discussion
15. Also there are many spelling mistakes and typos. All these are to be addressed.
Author Response

(The authors gave the same response as above.)
